# Quantification and Validation of an HPLC Method for Low Concentrations of Apigenin-7-*O*-Glucuronide in *Agrimonia pilosa* Aqueous Ethanol Extract Topical Cream by Liquid–Liquid Extraction

**DOI:** 10.3390/molecules28020713

**Published:** 2023-01-11

**Authors:** Jin Seok Lee, Yu Ran Nam, Hyun Jong Kim, Woo Kyung Kim

**Affiliations:** 1Department of Internal Medicine, Graduate School of Medicine, Dongguk University, 27 Dongguk-ro, Goyang 10326, Republic of Korea; 2Department of Physiology, College of Medicine, Dongguk University, 123 Dongdae-ro, Gyeongju 38066, Republic of Korea; 3Channelopathy Research Center (CRC), College of Medicine, Dongguk University, 32 Dongguk-ro, Goyang 10326, Republic of Korea

**Keywords:** *Agrimonia pilosa*, apigenin-7-*O*-glucuronide, cream, HPLC-DAD, validation

## Abstract

In this study, we aimed to develop and validate a pretreatment method for separating and analyzing the small amounts of biomarkers contained in topical cream formulations. Analyzing semisolid formulations that contain low concentrations of active ingredients is difficult. Cream formulations containing an aqueous ethanol extract of 0.1% Agrimonia pilosa is an example. Approximately 0.0013% of apigenin-7-*O*-glucuronide(A7OG) was contained as a biomarker in the cream. To determine the A7OG content present in the cream formulation, liquid–liquid extraction using dichlormethane was applied. In addition, the volume of the distribution liquid was measured using the peak ratios of the indicator component, A7OG, and an internal standard, baicalin. Subsequently, the A7OG content in the cream formulation was calculated. Using this time-saving method, A7OG can be simply analyzed without additional pretreatment steps, such as evaporation and reconstitution. Moreover, the validation results confirmed that this analytical method met all of the criteria. Consequently, A7OG was successfully isolated from the cream, analyzed, and quantified using the developed method.

## 1. Introduction

*Agrimonia Pilosa*, which belongs to the Rosaceae family, is a herb indigenous to the mountains and fields of Asia and Europe [1,2]. Additionally, this herb contains various tannins, such as agrimoniin and ellagic acid, and flavonoids, such as apigenin-7-*O*-glucuronide (A7OG) and luteolin-7-*O*-glucuronide. This herb, which has a slightly bitter and astringent taste, has been traditionally used to treat hemostasis, intestinal disorders, and inflammation [3,4]. Recent studies have reported the herb as having anticancer, antioxidant, and anti-inflammatory properties [5,6,7,8]. Therefore, it has emerged as a potential candidate for a new class of drugs with a wide range of benefits. As an efficacious natural resource, it is currently being researched and developed in both the cosmetic and pharmaceutical fields [9,10]. However, the quality control of herbal products is more challenging than that of synthetic chemical products. For example, identifying biomarkers that are stable and safe over long periods can be difficult. In addition, some biomarkers are present in trace amounts and may be subject to interference from various components, thereby hindering instrumental quantification [11,12,13,14]. Therefore, a growing demand exists for a quality control protocol for herbal medicines and cosmetics. Quality control can be achieved for the management of biomarkers and the evaluation of stability. There are ongoing studies to analyze and quantify these products, and many methods use high-performance liquid chromatography diode-array detector (HPLC-DAD) after increasing the sensitivity through sample pretreatment methods to remove the matrix [15,16,17]. Among the various techniques reported, liquid–liquid extraction (LLE) is a simple and economical method in which components are moved from one phase to another such that the two phases are not miscible with each other. This method enables the separation of unwanted matrices and other components. However, this process has certain limitations; for instance, a large volume of solvent is required to separate the matrix and the component. Moreover, this method may require several distribution processes. Although LLE can be easily used for simple component analyses, its application is difficult for quantitative purposes. Because the volume of the solvent in which the desired component is distributed cannot be accurately determined, additional preprocessing steps, including evaporation and reconstruction, are required. The more complex the process, the more time consuming, and a difference in the results between people can occur [18,19,20,21,22]. Therefore, to quantify the components present in the matrix, it is necessary to apply other pretreatments or to optimize the liquid–liquid extraction process as much as possible. In this study, a method of separating the unnecessary ingredients and components to be analyzed using liquid–liquid extraction with a specific organic solvent was investigated. LLE was used for the separation of A7OG from the cream matrix, because it is an easy and quick method. In addition, the volume of the separated solution containing A7OG was determined using baicalin, an internal standard substance showing the same distribution pattern as A7OG. This standard was also used to calculate the content of A7OG present in the cream formulation. The content of A7OG (chemical structure shown in Figure 1) in the previously prepared 0.1% *Agrimonia pilosa* aqueous ethanol extract (AE) was approximately 1.3%. Therefore, the cream formulation containing 0.1% AE contained approximately 0.0013% of A7OG. Therefore, the 0.1% AE cream contained 0.0013% A7OG, which is a considerably low concentration to be detected via DAD. In order for the 0.1% AE cream to be used as an investigational drug, it is necessary to be able to analyze the biomarker since strict quality control is required. Consequently, we aimed to establish and validate a liquid–liquid extraction-based HPLC-DAD method for analyzing and quantify the cream.

## 2. Results

### 2.1. Optimization of the Sample Preparation

When the 0.1% AE cream was dissolved in acetonitrile or methanol, the A7OG peak was barely observed. The concentration of A7OG in the diluted cream was approximately 2.4 μg/mL (assuming that 1 g of cream is 1 mL), which is higher than the LOQ of 0.873 μg/mL. Therefore, we assumed that if the concentration of A7OG was lowered, the detection of A7OG would be inhibited due to the matrix interference effect of the cream base. A peak that could not be quantified was confirmed even when the theoretical concentration was higher than the LOQ. Moreover, reducing the amount of diluting solvent makes it difficult to dissolve the cream. Even if the dilution factor is lowered, it is difficult to observe a peak with a sensitivity close to the LOQ. Therefore, the dilution of the 0.1% AE cream was not suitable for this experiment. In addition, the liquid–liquid extraction using organic solvents, such as acetone, hexane, and dichloromethane, was employed as an alternative method. As a result, the A7OG peak was observed only in dichloromethane; for the other solvents, the A7OG peak was barely detectable. Therefore, subsequent experiments were carried out by employing liquid–liquid extraction with dichloromethane. The analyzed chromatograms are shown in Figure 2.

### 2.2. Summary of the Validation

The validation results are summarized in Table 1.

### 2.3. System Suitability

A system suitability test was performed to evaluate whether the analysis system consisting of equipment worked as intended. The concentration of the standard solution corresponded to 100% of the A7OG target concentration in the 0.1% AE cream pretreatment solution. The solution was injected six times (the concentration of A7OG was 17.0 μg/mL and that of baicalin was 30 μg/mL). Consequently, the peak average area for A7OG was 413,138.3 and the percentage of the relative standard deviation (%RSD) was 0.4%. The peak average area and %RSD for baicalin were 323,756.5 and 0.2%, respectively.

### 2.4. Specificity

A blank matrix cream pretreatment solution and the 0.1% AE cream pretreated with the internal standard were analyzed to compare the obtained peak patterns at 335 nm (Figure 3). In the chromatogram, the retention time was 19.8 min for A7OG and 24.7 min for baicalin. The ultraviolet-visible (UV-vis) spectra of A7OG and baicalin were also compared in the standard and sample (Figure 4).

### 2.5. Linearity, Limit of Detection (LOD), and Limit of Quantitation (LOQ)

Three sets of solutions of 60, 80, 100, 120, and 140% target concentrations were prepared and analyzed. The linearity was examined with the concentration of the solution plotted on the *x*-axis and the peak area of the component plotted on the *y*-axis. The mean slope (S) was 25,289 and the mean intercept was 2206. As a result, the coefficient of determination (R2) of each calibration curve was higher than 0.995. Therefore, the linearity was confirmed within the concentration range. The average peak area ratio of the A7OG/internal standard was 1.27.

The LOD was 0.288 μg/mL, and the LOQ was 0.873 μg/mL.

### 2.6. Accuracy

As described in the sample preparation, three concentrations (60, 100 and 140%) of samples prepared in triplicate via a pretreatment process containing A7OG and internal standards in the blank cream, as well as the separation with dichloromethane, were analyzed and quantified. We compared the content of the sample with the theoretical content contained in the cream using the quantitative equation provided in Section 4.8. At a 60% concentration, the average recovery rate was 97% and the %RSD was 0.3%, whereas at a 100% concentration, the average recovery rate was 99.8% and the %RD was 0.5%. Further, the average recovery rate was 102% and the %RD was 0.4% at a 140% concentration.

### 2.7. Precision (Repeatability)

First, when six standard solutions with a 100% concentration were analyzed, the average ratio of the peak area of A7OG to the peak area of the internal standard was 1.28 and the %RSD was 0.56%. The theoretical value of the concentration of the A7OG obtained by analyzing the standard solution was 13.9 μg/g. This value represents the A7OG content in the cream converted inversely based on a standard solution of a 100% target concentration (17 μg/mL). The average recovery rate of the sample solution was 101.4%, and the %RSD was 0.20%.

### 2.8. Intermediate Precision

The intermediate precision in the laboratory was analyzed by different testers applying different test equipment over different test days. The experiment was conducted in the same way as described in Section 2.7. When six standard solutions with a 100% concentration were analyzed, the average ratio of the peak area of the A7OG peak to the peak area of the internal standard was 1.3 and the %RSD was 1.09%. The average recovery rate of the sample solution was 100.5%, and the %RSD was 0.15%.

### 2.9. Solution Stability

The 17 μg/mL concentration of six samples were stored at a room temperature of approximately 15 °C for approximately 43 h and then analyzed and compared with the initial analysis value. The average change rate of the peak area of the A7OG was −0.6%.

### 2.10. Quantification of the 0.1% AE Cream

The content of A7OG in the 0.1% AE cream in each of the three batches was determined, and an average content value of 12.2 μg/g and %RSD of 1.1% were obtained. Since the A7OG content in the *Agrimonia pilosa* extract was known in advance, the theoretical value was obtained as 12.7 μg/g. The actual value was approximately 4% lower than the theoretical value. The dichlormethane layer obtained in the pretreatment process was also analyzed, and the peak of the A7OG did not appear. Therefore, we assumed that the manufacturing process affected the content of A7OG. However, this aspect needs to be researched further. The analyzed concentrations were within the range of linearity verified values through validation. Therefore, it was confirmed that the A7OG in the 0.1% AE cream could be quantified by this method.

## 3. Discussion

Generally, cream formulations are analyzed after dissolving the cream in an organic solvent. In such cases, the dissolved cream must contain a sufficient amount of the analyte for HPLC-DAD analysis. Other methods should be considered when interference from the base matrix may occur or when the concentration of the analyte itself is low. In order to analyze the components that are present in the finished product, these components have to be separated from the matrix. This separation can be carried out using liquid–liquid extraction. The reason for adopting this method is that most of the cream bases are nonpolar and can distribute the water and index components present in the cream into two immiscible layers. Liquid–liquid extraction is a very simple and economical method; however, this method can be difficult to quantify, because it is difficult to determine the volume of solvent. In general, a method for volatilizing the solvent in which the components are dissolved is required, and another solvent can be subsequently added to dissolve them again. However, this method takes time and labor, and the reproducibility of the results depends on the skill level of the person performing the experiment. Therefore, in this study, the optimal solvent for separating A7OG from the cream base was selected, and an internal standard was used. Moreover, we found a suitable solvent for liquid–liquid extraction. We simultaneously determined the volume of the solution in which the A7OG was dissolved using an internal standard, bacicalin. Furthermore, we successfully measured the A7OG content in the 0.1% AE cream formulation. The target compound, A7OG, and other materials, such as cetanol and stearyl alcohol, were separated using dichloromethane. When the cream was mixed with dichloromethane, and the solution was separated, A7OG was distributed in the supernatant. The volume of the distributed supernatant and the concentration of A7OG can be calculated using an internal standard material with the same distribution pattern as that of A7OG. Subsequently, this analytical method was validated, and all of the validation criteria were met. In addition, this method was verified once more by quantifying the commercially produced 0.1% AE cream.

It is becoming increasingly important to perform quality control and consistently manage various products, such as pharmaceuticals and cosmetics. In addition, there are many commercial topical products, such as cosmetics and medicines, which contain natural product extracts. However, it is difficult to develop innovative, simple, time saving, and reproducible test methods for natural product components. In this respect, the methods developed this time can be helpful for related practitioners.

Moreover, we further plan to conduct a long-term stability test and quality control of this cream by applying the technique developed in this study. In addition, investigations for developing an optimal separation technology suitable for the characterization of formulations and compound will be undertaken.

## 4. Materials and Methods

### 4.1. Chemicals and Reagents

The A7OG standard and baicalin were purchased from Biopurify (Chengdu, China) and Alladin-e (Shanghai, China), respectively. The acetonitrile, methanol, and water were HPLC-grade. Extra-pure grade dichloromethane (J.T.Baker) and phosphoric acid (Daejung, Korea) were used in this experiment.

### 4.2. Preparation of the Agrimonia pilosa Extract

The *Agrimonia pilosa* herb (10 kg) was added to 70% ethanol at a weight ratio of 20 times (200 L). The resultant mixture was refluxed at 90 °C for 4 h. Subsequently, the mixture was filtered and concentrated under reduced pressure at approximately 40 °C. Finally, the concentrate was lyophilized at −80 °C for approximately one day to obtain a powder. The yield was approximately 10%. Moreover, the content of A7OG in this extract was approximately 12.6 mg/g.

### 4.3. Preparation of 0.1% AE Cream

The extract cream consisted of AE (1% by weight), water, propylene glycol, liquid paraffin, cetanol, stearyl alcohol, polysorbate 60, monostearate sorbitan, benzyl alcohol (preservative), and citric acid hydrate. First, the *Agrimonia pilosa* herbs were extracted using water with ethanol, freeze-dried, and powdered. The water phase was prepared by mixing purified water and citric acid. The oil phase was then prepared by stirring and dissolving the remaining ingredients, including the extract, at approximately 90 °C. Subsequently, the water and oil phases were mixed and stirred uniformly under heat. Thereafter, the heat was rapidly lowered to prepare a cream formulation.

### 4.4. Instrument and Chromatographic Condition

The HPLC system with a Shimadzu SPD-M20A photodiode array detector (Shimadzu, Kyoto, Japan) was used for the analysis and validation. The Waters e2695 alliance (Waters Corporation, Milford, MA, USA) HPLC system was used for the inter-laboratory validation. The temperature of the column (YoungJin Biochrom INNO-P C18 5 µm, 4.6 × 150 mm) was set at 30 °C, and the flow rate was maintained at 1 mL/min. The injection volume was 10 µL, and the detection wavelength was 335 nm. The mobile phase consisted of (A) distilled water with 0.1% phosphoric acid and (B) acetonitrile. For the separation of A7OG and baicalin, the following gradient was used: solution B was changed from 90% to 75% between 0 and 30 min; then, B was 75% to 90%, from 30 to 40 min, and isocratic from 40 to 50 min.

### 4.5. Optimization of the Sample Preparation Method

To develop an appropriate pretreatment method, the following experiments were conducted. The first method was to take five grams of 0.1% AE cream and dissolve it in 20 mL of acetonitrile or methanol. After being dissolved, it was filtered and analyzed by HPLC-DAD. For the second method, liquid–liquid extraction was performed. Five grams of the 0.1% AE cream was weighed, and 20 mL of organic solvent was added to it. Examples of organic solvents that were included: acetone, hexane, dichloromethane, and ethyl acetate. When the cream was divided into two layers, the layer that was immiscible with the organic solvent was taken and analyzed after being filtered.

### 4.6. Sample Preparation for the Validation

Seventeen milligrams of the A7OG standard were dissolved in methanol in a 100 mL volumetric flask to create the standard stock solution (Solution A). Secondly, 25 mg of baicalin standard was dissolved in methanol and adjusted to 100 mL in a volumetric flask as an internal standard solution. Furthermore, 6, 8, 10, 12 and 14 mL of the standard stock solution were mixed with 12 mL of the internal standard solution, and water was added to result in exactly 100 mL. These solutions were used as the standard solutions (concentrations of 10.2, 13.6, 17.0, 20.4 and 23.8 μg/mL, respectively). The concentration of the internal standard in the standard solutions was 30 μg/mL.

For validating the accuracy and precision, blank matrix sample preparation was also needed. Five grams of blank matrix cream were weighed; placed in a 100 mL volumetric flask; and mixed with 20 mL of dichloromethane, 500 µL of the internal standard solution, and 0.3, 0.5 and 0.7 mL of solution A as the standard stock solution (60, 100 and 140% of the target concentration, respectively). After being sufficiently dissolved, the supernatant was collected and centrifuged at 13,500 rpm for 10 min. Once clear, the supernatant was filtered and used for quantification using the HPLC system.

### 4.7. Validation

In this study, system suitability, specificity, linearity, accuracy, precision, and robustness (stability) were validated for the assay. A blank matrix of the cream formulation was used to validate the specificity, accuracy, precision, and solution stability. The criteria for the validation test were followed by considering the A7OG concentration contained in the formulation: “AOAC Guideline for Single Laboratory Validation of Chemical Methods for Dietary Supplements and Botanicals” [23]. As the A7OG content unit in the formulation was μg/g, the recovery rate and relative standard deviation range of the precision and accuracy were set within 90~110%, and the relative deviation of the peak area ratio of the system suitability was set within 10%. In addition, the limit of detection and limit of quantitation were calculated according to the following formula.

Each calculation formula is as follows:Limit of Detection = 3.3 × σ/S,
Limit of Quantitation = 10 × σ/S,
where S is the mean of the slopes, and σ is the standard deviation of the intercepts.

### 4.8. Quantification of the 0.1%AE Cream

The following procedure was followed to quantify the A7OG content in the previously prepared 0.1% cream. Please refer to Section 4.6 to prepare the following solutions. Seventeen milligrams of the A7OG standard was dissolved in methanol in a 100 mL volumetric flask (standard stock solution). Subsequently, twenty-five milligrams of baicalin standard were dissolved in methanol and adjusted to 100 mL in a volumetric flask (internal standard solution). Next, 10 mL of the standard stock solution was mixed with 12 mL of the internal standard solution, and the resultant solution was diluted with water to result in a volume that was exactly 100 mL. These solutions were used as the standard solutions. The concentrations of the A7OG and baicalin were 17 and 30 μg/mL, respectively.

Five grams of the 0.1% AE cream were mixed with 20 mL of dichloromethane and 500 µL of the internal standard solution. After sufficiently dissolving the cream, the supernatant was taken aside and filtered before the analysis. The equation for obtaining the content of the A7OG in the 0.1% AE cream is given below.

The quantitative equation is given by:$$=\frac{\left(5\times WI.S\times S.s\times Rc\right)}{\left(Rs\times S.I\times W\right)}$$*WI.S:* Weight of the internal standard (i.e., baicalin);*S.s:* The concentration of A7OG in the standard solution (concentration correction required according to the purity of the standard product);*Rc:* The ratio (A7OG peak area/internal standard peak area) in the sample solution;*Rs:* The ratio (A7OG peak area/internal standard peak area) of the standard solution;*S.I:* The concentration of the internal standard in the standard solution (concentration correction required according to the purity of the standard product);*W:* Weight of the cream.

## 5. Conclusions

In this study, a method for quantifying the A7OG content in a natural *Agrimonia pilosa* extract was successfully optimized and verified. Liquid–liquid extraction was performed using a dichloromethane solution to separate A7OG, a marker component, for analysis. In addition, the A7OG content in the cream formulation could be measured by using the peak area ratio of the A7OG and the internal standard. Since this method did not require additional pretreatment, it was possible to measure the A7OG content simply and accurately. The results obtained from this experiment indicate that the developed method is suitable for the quality control of topical cream formulations.

## Figures and Tables

**Figure 1 molecules-28-00713-f001:**
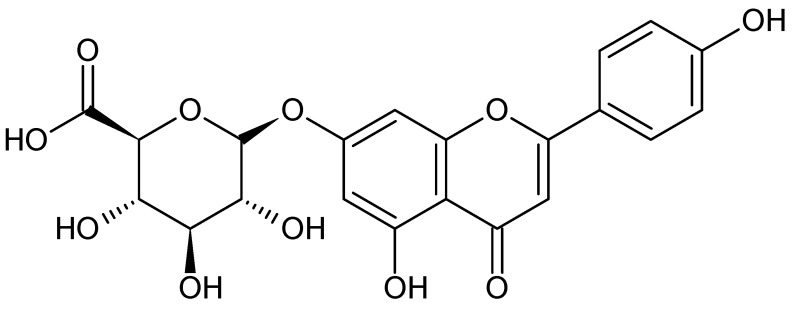
Structure of apigenin-7-*O*-glucuronide.

**Figure 2 molecules-28-00713-f002:**
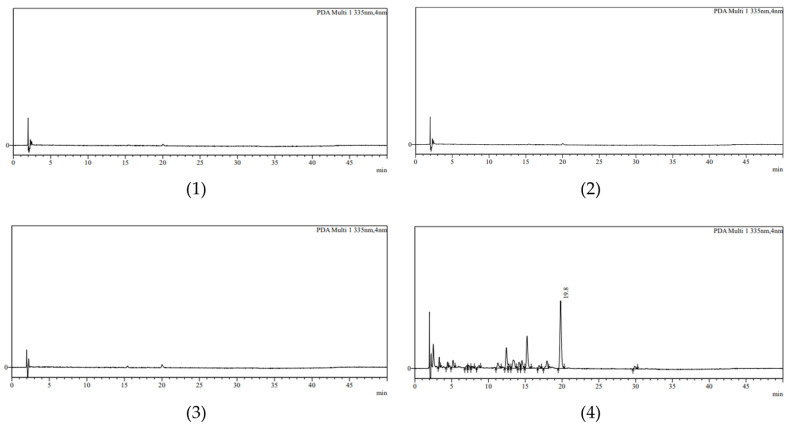
HPLC-DAD chromatogram according to the pretreatment method: (**1**) 0.1% AE cream dissolved in acetonitrile; (**2**) analysis after liquid–liquid extraction using acetone; (**3**) analysis after liquid–liquid extraction using hexane; (**4**) analysis after liquid–liquid extraction using dichloromethane.

**Figure 3 molecules-28-00713-f003:**
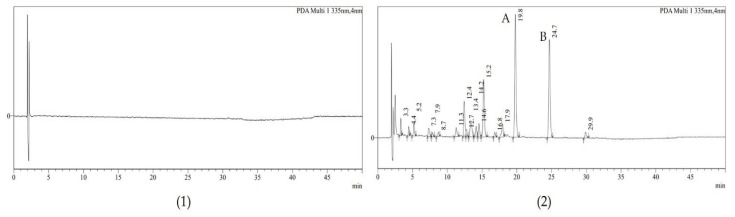
HPLC-DAD chromatogram of the (**1**) blank matrix cream; (**2**) 0.1% AE cream pretreated solution with internal standard. Peak A represents A7OG, and peak B represents baicalin, which is the internal standard.

**Figure 4 molecules-28-00713-f004:**
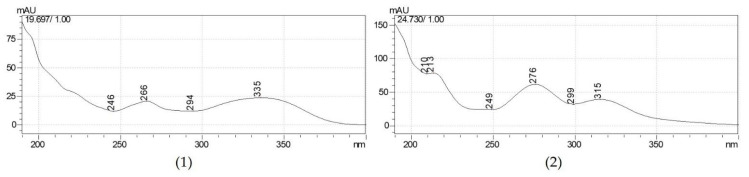
UV-vis spectrum of (**1**) A7OG and (**2**) baicalin from the 0.1% AE cream pretreated solution with internal standard and standard solution with internal standard.

**Table 1 molecules-28-00713-t001:** System suitability.

Parameter		Result	
System Suitability		%RSD of A7OG AREA	0.4%
		%RSD of Baicalin AREA	0.2%
Linearity (μg/mL)		10.2–23.8	
Slope		25,289	
Intercept		−9184	
Regression coefficient (R^2^)		0.9999	
%Recovery	60%	97 ± 0.3	
	100%	99.8 ± 0.5	
	140%	102 ± 0.4	
Precision (%RSD)		0.20%	
Intermediate Precision (%RSD)		0.15%	
LOD (μg/mL)		0.28	
LOQ (μg/mL)		0.87	

## Data Availability

The data that support this study are available from the corresponding author upon reasonable request.

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
