# Peer review of "Quantification and Validation of an HPLC Method for Low Concentrations of Apigenin-7-*O*-Glucuronide in *Agrimonia pilosa* Aqueous Ethanol Extract Topical Cream by Liquid–Liquid Extraction"

_molecules, 2023, doi:10.3390/molecules28020713_

Round 1
Reviewer 1 Report
In several parts of the text, it is commented that the concentration in the test solutions are a percentage of the target concentration; however, at no time is the concentration of A7OG in the Agrimonia pilosa herbs extract or its concentration in the 1% AE cream indicated. It is necessary to know this value to verify if the concentration of A7OG in the solutions of the sample in methanol and acetonitrile is below the detection limit, as indicated by the authors. Taking into account the concentration data provided, it does not seem that this is the reason why the compound is not detected in said solutions. The A7OG concentration in solutions assayed should be included and a comment should be made about it.
3.2. System suitability. The concentration of the solutions used in the study must be indicated. This assay should be performed at different concentration levels, specially at concentrations close to the LOQ. The results of this study show a similar reproducibility in the area means obtained for the A7OG pattern and for the A7OG/I.S. ratio, so it does not seem necessary to use an internal pattern in the measure. Nothing is said about this fact in the results and discussion sections.
3.4. Linearity, LOD and LOQ. In the LOD and LOQ study, the authors indicate that the calibration line is made with a set of solutions with concentrations between 60% and 140% of the target concentration. It should be specified in the text if these solutions refer to those prepared in water (section 2.5) and if they have been subjected to the extraction process or not. On the other hand, if the minimum concentration tested has a concentration of 10.2 µg/mL, solutions close to the LOQ (0.8 µg/mL) should be analyzed to confirm said parameter. Is this limit of quantification sufficient to analyse commercial samples containing Agrimonia pilosa extract?
3.5. Accuracy. The number of replicates carried out at each concentration level analyzed must be indicated.
3.8 Solution stability. The concentration of the solutions used must be indicated.
To complete the validation of the method, commercial samples containing Agrimonia pilosa extract should be analysed.
Author Response
We thank you for your thoughtful suggestions and insights, which have enriched the manuscript and produced a better and more balanced account of the research.
Reviewer 1
In several parts of the text, it is commented that the concentration in the test solutions are a percentage of the target concentration; however, at no time is the concentration of A7OG in the Agrimonia pilosa herbs extract or its concentration in the 1% AE cream indicated. It is necessary to know this value to verify if the concentration of A7OG in the solutions of the sample in methanol and acetonitrile is below the detection limit, as indicated by the authors. Taking into account the concentration data provided, it does not seem that this is the reason why the compound is not detected in said solutions. The A7OG concentration in solutions assayed should be included and a comment should be made about it.
We have made changes in our manuscript as per your suggestions. We conducted the necessary experiment, and the file has been attached herewith. Moreover, 1 μg/ml (repeated 6 times) and 0.25μg/ml solutions of A7OG and a solution of 0.1% AE cream in methanol were analyzed in the same way as described in the paper.
We agree with you, in that the concentration of A7OG in the herb extract and that in the 0.1% AE cream should have been mentioned. The concentration of A7OG in the agrimony extract we used was approximately 12.6 mg/g, which corresponds to about 1.3% in the extract. Moreover, since the cream contained 0.1% of the Agrimonia pilosa extract, the theoretical value of the content of A7OG in the cream was 12.6 μg/g.
If the 0.1% AE cream is diluted with methanol or acetonitrile as in this experiment (assuming 1 g of cream is 1 mL), the concentration of A7OG would be approximately 2.4 μg/ml, which is higher than the LOQ. Therefore, we agree that it should be quantified before the experiment. However, the size of the actual peak is smaller than that obtained by analyzing 1 μg/ml of A7OG. By enlarging chromatogram, we can observe peak that can be identified but not quantified. Assuming that the concentration analyzed by diluting the cream is approximately 0.2 μg/ml, the dilution factor is 10 times higher than the theoretical value of 2.4 μg/ml. Therefore, it is difficult to observe this as a laboratory error. Hence, we concluded that the detection of A7OG, a natural indicator component, is difficult due to the matrix of the base present in the cream.
3.2. System suitability. The concentration of the solutions used in the study must be indicated. This assay should be performed at different concentration levels, specially at concentrations close to the LOQ. The results of this study show a similar reproducibility in the area means obtained for the A7OG pattern and for the A7OG/I.S. ratio, so it does not seem necessary to use an internal pattern in the measure. Nothing is said about this fact in the results and discussion sections.
The concentration of the solution used for system suitability was specifically noted. The concentration of A7OG was 17.0 μg/mL and that of baicalin as an internal standard was 30 μg/mL. This concentration of A7OG corresponds to the close concentration value of the 0.1% AE cream pre-treatment liquid. The system suitability experiment was not performed at a value close to the LOQ because the different A7OG concentrations in the 0.1% AE cream were quality controlled. Moreover, as you have mentioned, each reproducibility of the A7OG and internal standard peak is similar and meets the criteria therefore, I agree that it is meaningless to set a standard up to that ratio. The ratio of the A7OG and internal standard peak areas is an indicator for quantifying the 0.1% AE cream. This information was moved to the quantification section 3.10. In addition, the system suitability results were described in a paragraph instead of a table, and the entire validation results table was added in section 3.1, as suggested by Reviewer 2.
3.4. Linearity, LOD and LOQ. In the LOD and LOQ study, the authors indicate that the calibration line is made with a set of solutions with concentrations between 60% and 140% of the target concentration. It should be specified in the text if these solutions refer to those prepared in water (section 2.5) and if they have been subjected to the extraction process or not. On the other hand, if the minimum concentration tested has a concentration of 10.2 µg/mL, solutions close to the LOQ (0.8 µg/mL) should be analyzed to confirm said parameter. Is this limit of quantification sufficient to analyse commercial samples containing Agrimonia pilosa extract?
We have made corrections in the manuscript as per your suggestion. It was sepcified as Solution A to clarify the contents(Section 2.6.). The title for section 2.6 has been modified as “Sample preparation for validation” and a section quantifying the 0.1% AE cream has also been added.
A 1 μg/ml solution close to the LOQ was repeatedly analyzed, and the results are attached herewith. Moreover, the value near LOD (0.25 μg/ml) was analyzed. An AE cream with a concentration of 0.01% will also be analyzable. However, we did not manufacture the AE cream with the 0.01% concentration because it is difficult to show the desired pharmacological effect using this concentration.
This LOQ is sufficient to analyze commercial samples. In this experiment, the content of A7OG in the 0.1% AE cream is approximately 12.2 μg/g, and this concentration corresponds to the average value of the analysis of Agrimonia pilosa from various places before drug design. Therefore, commercially produced creams can also be quality controlled according to the same standard.
3.5. Accuracy. The number of replicates carried out at each concentration level analyzed must be indicated.
Thank you for pointing this out. The number of replicates carried out at each concentration have been mentioned in the manuscript (Section 3.6. triplicate each concentration level).
3.8 Solution stability. The concentration of the solutions used must be indicated.
The concentration of the solutions used in this study have been indicated (17 μg/ml concentration of six samples).
To complete the validation of the method, commercial samples containing Agrimonia pilosa
The 0.1% AE cream was quantified, and this process has been described in section 2.8. This cream can be sold as general product and will be used further for clinical trials. We are currently conducting a stability test to develop this cream as a medicine, and this test will be applied for the same.
Reviewer 2 Report
The manuscript entitled “Quantification and validation of an HPLC method for low concentrations of apigenin-7-O-glucuronide in Agrimonia pilosa aqueous ethanol extract topical cream by liquid-liquid extraction”, authored Lee et al., deals with developing a new quality control protocol for herbal plant material. The topic of this manuscript is important and current, and results could be interesting for readers. However, some changes have to be entered into the revised version of the manuscript before it can be further processed:
1. more information about the liquid-liquid extraction technique in the introduction should be added
2. in chapter 2.2 more precisely information about extract preparation should be added, such as alcohol percentage in the extraction mixture, extraction temperature and time, freeze-drying process parameters, powder formation method.
3. equations 1, 2 and following should be placed in the methodology section, not in results.
4. please provide all validation parameters in tabular form
5. there is no indication of innovations in research and application in practice. Please complete with e.g. dissolution test of active compound from cream using the described method.
6. I also miss reference to other studies, is this technique already used in practice? What is the effect of using it?
Author Response
We thank you for your thoughtful suggestions and insights, which have enriched the manuscript and produced a better and more balanced account of the research.
Reviewer 2
The manuscript entitled “Quantification and validation of an HPLC method for low concentrations of apigenin-7-O-glucuronide in Agrimonia pilosa aqueous ethanol extract topical cream by liquid-liquid extraction”, authored Lee et al., deals with developing a new quality control protocol for herbal plant material. The topic of this manuscript is important and current, and results could be interesting for readers. However, some changes have to be entered into the revised version of the manuscript before it can be further processed:
Thank you for your insightful comments. Thanks to this, I was able to improve the quality of the thesis and at the same time examine the experiment in depth.
- more information about the liquid-liquid extraction technique in the introduction should be added
As per your suggestion, we have added some information regarding liquid-liquid extraction in the introduction. We have described the advantages and disadvantages of liquid-liquid extraction and made the purpose of our experiment clearer.
- in chapter 2.2 more precisely information about extract preparation should be added, such as alcohol percentage in the extraction mixture, extraction temperature and time, freeze-drying process parameters, powder formation method.
Thank you for bringing this to our attention. We have added a section on the Agrimonia pilosa extraction process.
- equations 1, 2 and following should be placed in the methodology section, not in results.
As per your suggestion, equations 1 and 2 were moved to the methodology section.
- please provide all validation parameters in tabular form
These parameters have been included in Table 1.
- there is no indication of innovations in research and application in practice. Please complete with e.g. dissolution test of active compound from cream using the described method.
This time-saving method is relatively simple and very useful in terms of quality control. Because there is no need for other process such as evaporation, just analyze immediately after liquid liquid extraction. It’s a rapid and reproducible method for semi-solid formulations containing trace amounts of natural substances.
Liquid-liquid extraction is a widely used method, and dichloromethane is often used as a solvent in this method. However, it is difficult to quantify the desired component using liquid-liquid extraction. This is because a large amount of solvent may be required to dispense the desired component. Moreover, since it is difficult to determine the amount of solvent in which the active ingredient is dissolved, the component content cannot be quantified. In this case, most solutions have to be evaporated and then diluted again. This process it is time and labor consuming. The more complex the process, the less reproducible it is, and the results tend to vary depending on the skill level of the person performing the experiment. Therefore, we developed a rapid and consistent quantification method using a single application of LLE and an internal standard. In this method, the desired component was separated using dichloromethane. The volume of the distributed supernatant and concentration of A7OG in the cream were calculated using an internal standard substance, Baicalin, which shows the same distribution pattern as A7OG.
- I also miss reference to other studies, is this technique already used in practice? What is the effect of using it?
Yes, this technique is currently being used to conduct quality control and stability tests for 0.1% AE cream in a clinical trial. Moreover, this method is also being used to set the expiration date of medicines. Pharmaceutical research institutes with GMP standards are also using this technique by receiving it from us. They report that this technique is simple and novel.
Round 2
Reviewer 2 Report
Thank you for the changes made. I accept the manuscript in present form.